# HQ-ISNet: High-Quality Instance Segmentation for Remote Sensing Imagery

**Hao Su [1], Shunjun Wei [1,\*], Shan Liu [1], Jiadian Liang [1], Chen Wang [1], Jun Shi [1] and Xiaoling Zhang [1]**

School of Information and Communication Engineering, University of Electronic Science and Technology of China, Chengdu 611731, China

\* Correspondence: suhao@std.uestc.edu.cn

**Abstract:** Instance segmentation in high-resolution (HR) remote sensing imagery is one of the most challenging tasks and is more difficult than object detection and semantic segmentation tasks. It aims to predict class labels and pixel-wise instance masks to locate instances in an image. However, there are rare methods currently suitable for instance segmentation in the HR remote sensing images. Meanwhile, it is more difficult to implement instance segmentation due to the complex background of remote sensing images. In this article, a novel instance segmentation approach of HR remote sensing imagery based on Cascade Mask R-CNN is proposed, which is called a high-quality instance segmentation network (HQ-ISNet). In this scheme, the HQ-ISNet exploits a HR feature pyramid network (HRFPN) to fully utilize multi-level feature maps and maintain HR feature maps for remote sensing images' instance segmentation. Next, to refine mask information flow between mask branches, the instance segmentation network version 2 (ISNetV2) is proposed to promote further improvements in mask prediction accuracy. Then, we construct a new, more challenging dataset based on the synthetic aperture radar (SAR) ship detection dataset (SSDD) and the Northwestern Polytechnical University very-high-resolution 10-class geospatial object detection dataset (NWPU VHR-10) for remote sensing images instance segmentation which can be used as a benchmark for evaluating instance segmentation algorithms in the high-resolution remote sensing images. Finally, extensive experimental analyses and comparisons on the SSDD and the NWPU VHR-10 dataset show that (1) the HRFPN makes the predicted instance masks more accurate, which can effectively enhance the instance segmentation performance of the high-resolution remote sensing imagery; (2) the ISNetV2 is effective and promotes further improvements in mask prediction accuracy; (3) our proposed framework HQ-ISNet is effective and more accurate for instance segmentation in the remote sensing imagery than the existing algorithms.

**Keywords:** instance segmentation; HRFPN; ISNetV2; SSDD; NWPU VHR-10; remote sensing images

## 1. Introduction

With the rapid development of imaging technology in the field of remote sensing, high-resolution (HR) remote sensing images are provided by many airborne and spaceborne sensors, for instance, RADARSAT-2, Gaofen-3, TerraSAR-X, Sentinel-1, Ziyuan-3, Gaofen-2 and unmanned aerial vehicles (UAV). Nowadays, these HR images have been applied to the national economy and the military fields, such as urban monitoring, ocean monitoring, maritime management, and traffic planning [1–3]. In particular, territories such as military precision strike and

maritime transport safety tend to take full advantage of the HR remote sensing images for object detection and segmentation [3–5].

Traditional object detection methods in remote sensing (RS) imagery mainly pay attention to the detection results with the bounding boxes and the rotational bounding boxes, as shown in Figure 1b,c. Cheng et al. [6] proposed an approach to improve the performance of target detection by learning the rotation-invariant CNN (RICNN) model. Ma et al. [7] applied the You Only Look Once (YOLOv3) approach to locate collapsed buildings from remote sensing images after the earthquake. Gong et al. [8] put forward a context-aware convolutional neural network (CA-CNN) method to improve the performance of object detection. Liu et al. [9] proposed a multi-layer abstraction saliency model for airport detection in synthetic aperture radar (SAR) images. Wei et al. [10] came up with a HR ship detection network (HR-SDNet) to perform precise and robust ship detection in SAR images. Deng et al. [11] devised a method to detect multiscale artificial targets in remote sensing images. An et al. [12] came up with a DRBox-v2 with rotatable boxes to boost the precision and recall rates of detection for object detection in HR SAR images. Xiao et al. [13] came up with a novel anchor generation algorithm to eliminate the deficiencies in the previous anchor-based detectors. However, these detection results with the bounding boxes and the rotational bounding boxes do not reflect the pixel-level contours of the original targets.

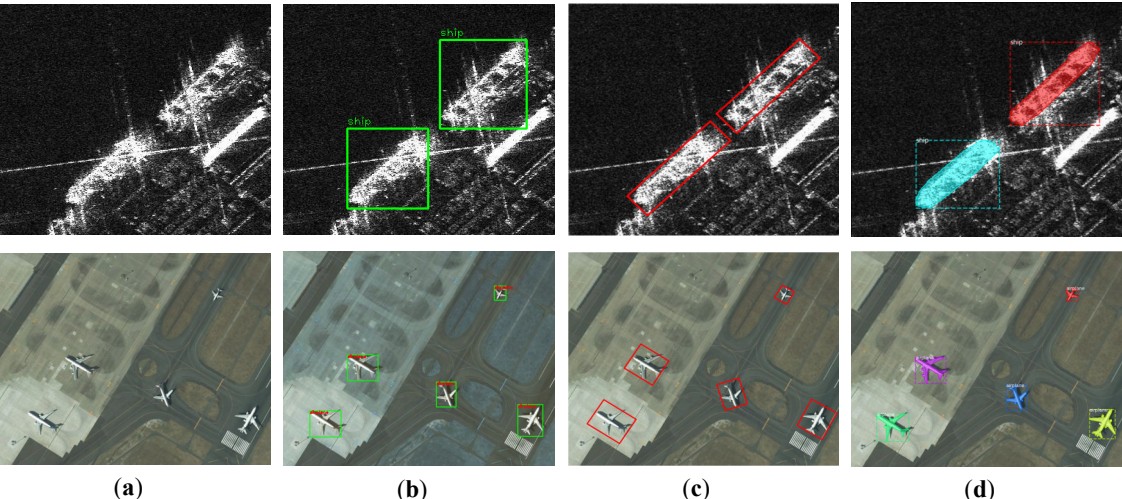

|       (**a**)       |       (**b**)       |       (**c**)       |       (**d**)       |

**Figure 1.** Examples of objects in the high-resolution (HR) remote sensing imagery. (**a**) original images; (**b**) bounding box results; (**c**) rotational bounding box results; (**d**) instance mask results.

Traditional semantic segmentation methods in remote sensing imagery mainly focus on pixel-level segmentation results. Shahzad et al. [14] used Fully Convolution Neural Networks to automatically detect man-made structures, especially buildings in very HR SAR Images. Chen et al. [15], based on a fully convolutional network (FCN), proposed a symmetrical dense-shortcut FCN (SDFCN) and a symmetrical normal-shortcut FCN (SNFCN) for the semantic segmentation of very HR remote sensing images. Yu et al. [16] came up with an end-to-end semantic segmentation framework that can simultaneously segment multiple ground objects from HR images. Peng et al. [17] came up with dense connection and FCN (DFCN) to automatically acquire fine-grained feature maps of semantic segmentation for HR remote-sensing images. Nogueira et al. [18] came up with a novel method based on ConvNets to accomplish semantic segmentation in HR remote sensing images. However, these segmentation results cannot distinguish different instances in each category. Therefore, instance segmentation is introduced into the field of remote sensing.

Instance segmentation in remote sensing (RS) images is a complicated problem and one of the most challenging tasks [3,19]. It aims to predict both the location and the semantic mask of each instance in an image, as shown in Figure 1d. This task is much harder than object detection and semantic segmentation. However, there are rare methods currently suitable for instance segmentation in RS images. Meanwhile, it is more difficult to implement instance segmentation on

HR RS images due to the complex background of remote sensing images. Therefore, this paper focuses on a high-quality instance segmentation method for remote sensing images, especially for high-resolution artificial targets.

Nowadays, many instance segmentation methods have emerged in the area of computer vision, which uses FPN structures as the backbone network, such as Mask R-CNN [19], Cascade Mask R-CNN [20,21], Mask Scoring R-CNN [22]. In the remote sensing field, Mou et al. [2] came up with a novel method to perform vehicle instance segmentation of aerial images and videos obtained by UAV. Su et al. [3] introduced the precise regions of interest (RoI) pooling into the Mask R-CNN to solve the problem of loss of accuracy due to the coordinate quantization in optical remote sensing images. However, these methods mostly utilize low-resolution representations or restore high-resolution representations for instance segmentation. Therefore, these methods are not appropriate for instance segmentation at the pixel-level in the HR RS images due to the huge loss of spatial resolution. Furthermore, in Cascade Mask R-CNN [20,21], the lack of interactive information flow between the mask branches will lead to the loss of the ability to gradually adjust and enhance between stages. In this article, a novel instance segmentation approach of HR remote sensing imagery based on Cascade Mask R-CNN [20,21] is proposed to address these problems, which we call the high-quality instance segmentation network (HQ-ISNet).

First, the HR feature pyramid network (HRFPN) is introduced into pixel-level instance segmentation in remote sensing images to fully utilize multi-level feature maps and maintain HR feature maps. Next, to refine mask information flow between mask branches, the instance segmentation network version 2 (ISNetV2) is proposed to promote further improvements in mask prediction accuracy. Then, we construct a new, more challenging dataset based on the synthetic aperture radar (SAR), ship detection dataset (SSDD) and the Northwestern Polytechnical University very-high-resolution 10-class geospatial object detection dataset (NWPU VHR-10) for remote sensing images' instance segmentation, which can be used as a benchmark for evaluating instance segmentation algorithms in the HR remote sensing images. Finally, the proposed HQ-ISNet is optimized in an end-to-end manner. Extensive experimental analyses and comparisons on the SSDD dataset [23] and the NWPU VHR-10 dataset [3,6] prove that the proposed framework is more efficient than the existing instance segmentation algorithms in the HR remote sensing images.

The main contributions of this article are shown below:

- We introduce HRFPN into remote sensing image instance segmentation to fully utilize multi-level feature maps and maintain HR feature maps, so as to solve the problem of spatial resolution loss in FPN;
- We design an ISNetV2 to refine mask information flow between mask branches, thereby promoting the improvement in mask prediction accuracy;
- We construct a new, more challenging dataset based on the SSDD and the NWPU VHR-10 dataset for remote sensing images instance segmentation, and it can be used as a benchmark for evaluating instance segmentation algorithms in the HR remote sensing images. In addition, we provide a study baseline for instance segmentation in remote sensing images;
- Most importantly, we are the first to perform instance segmentation in SAR images.

The organization of this paper is as follows. Section 2 is related to object detection and instance segmentation. Section 3 presents our instance segmentation approach. Section 4 describes the experiments, including the dataset description, evaluation metrics, experimental analysis, and experimental results. Section 5 discusses the impact of the dataset. Section 6 comes up with a conclusion.

## 2. Related Work

### 2.1. Object Detection

Object detection needs to both declare the existence of a target belonging to the specified category and locate it in the image with a bounding box. The existing object detectors can be roughly split into two categories. The one is one-stage object detectors, which can perform object detection

without proposals, such as YOLO v1-v3 [24–26], Single Shot MultiBox Detector (SSD) [27]. Fu et al. [28] put forward a Deconvolutional SSD (DSSD) for introducing additional context into the SSD to enhance detection performance. Li et al. [29] came up with Feature Fusion SSD (FSSD), which used a feature fusion module to enhance the detection performance. Lin et al. [30] put forward a RetinaNet, which utilized Focal Loss to address the class imbalance problem. The other is two-stage object detectors that generates proposals and then makes predictions for these proposals, such as Region with convolutional neural networks (R-CNN) [31], Fast R-CNN [32], Faster R-CNN [33]. Lin et al. [34] proposed a feature pyramid network (FPN) to utilize multi-level features. For high-quality object detection, Cai et al. [35] came up with a Cascade R-CNN, which consists of a series of detectors trained with increasing IoU thresholds. In short, compared with the one-stage detector, the two-stage detector has more accurate positioning and higher target recognition accuracy, but the one-stage detector has faster inference speed.

### 2.2. Instance Segmentation

Instance segmentation aims to predict both the location and the semantic mask of each instance in an image. This task is much harder than object detection and semantic segmentation. At present, the existing instance segmentation approaches can be summarily split into two categories: (1) detection-based methods first detect objects then perform segmentation within each bounding box. He et al. [19] came up with Mask R-CNN that adds a mask branch in parallel based on Faster R-CNN to predict instance masks at pixel-level. Mask R-CNN is shown in Figure 2. Liu et al. [36] put forward a novel approach, namely the Path Aggregation Network (PANet), to boost the information flow by adding a bottom-up path beyond FPN. Chen et al. [37] put forward MaskLab that utilized position-sensitive scores to acquire better segmentation results. Chen et al. [20] proposed a Hybrid Task Cascade to improve instance segmentation performance by adding semantic segmentation branches and training together with other branches. Huang et al. [22] came up with Mask Scoring R-CNN to address the problem of scoring masks to improve the quality of the predicted instance mask; (2) segmentation-based methods first obtain a pixel-level segmentation map in the entire image and then recognizes target instances. Liang et al. [38] came up with Proposal-Free Network (PFN) for instance-level object segmentation. Bai et al. [39] combined watershed algorithms and deep learning methods to generate image energy maps to perform instance segmentation.

In this paper, we follow the research line based on detection methods and further study the instance segmentation for remote sensing imagery.

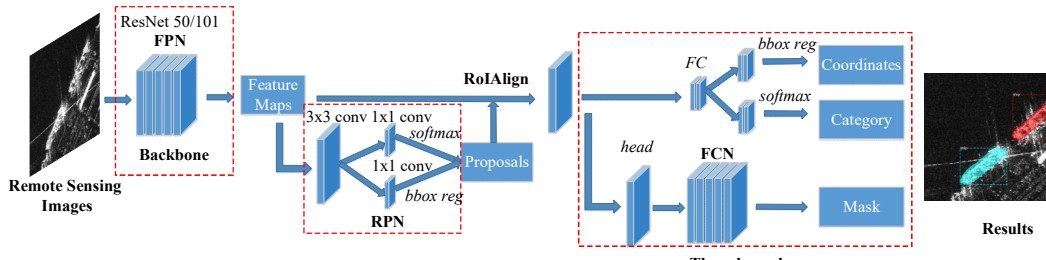

**Figure 2.** The architecture of the Mask R-CNN.

## 3. The Methods

The proposed network will be described in detail in this section.

### 3.1. Detailed Description of the HQ-ISNet

The framework of HQ-ISNet based on Cascade Mask R-CNN [21] is shown in Figure 3. First, an HR feature pyramid networks (HRFPN) replaces the original FPN to fully utilize multi-level feature maps; next, the candidate proposals are generated by the RPN; finally, an instance segmentation network version 2 (ISNetV2) is used to refine the original mask branches and is executed to obtain

the final instance segmentation results. In this section, we will present our proposed instance segmentation approach in detail.

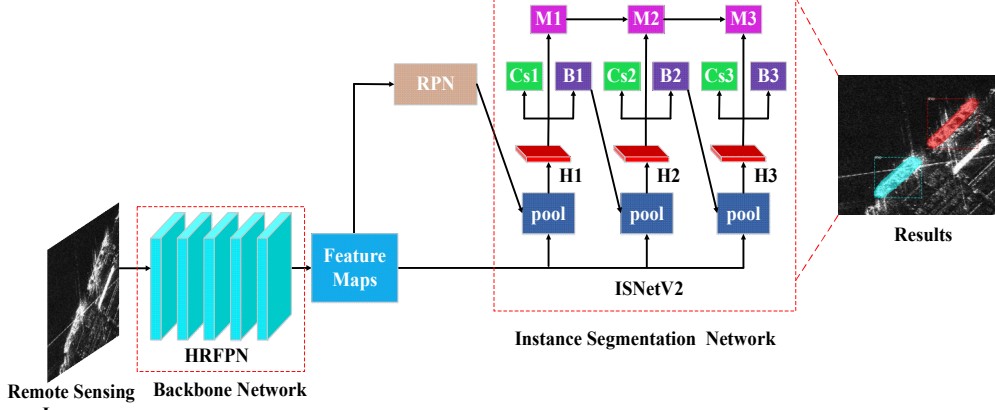

**Figure 3.** Illustration of the high-quality instance segmentation network (HQ-ISNet) approach where "HRFPN" indicates a backbone network; "RPN" indicates the proposals; "Cs" indicates the classification; "M" denotes the mask branch; "B" represents the bounding box; "H" denotes the detection head; "pool" means region feature extraction.

### 3.1.1. Backbone Network and RPN

Currently, most instance segmentation methods use FPN structures as the backbone network, such as Mask R-CNN [19]. However, these methods mostly utilize low-resolution representations or restore high-resolution representations for instance segmentation, resulting in a huge loss of spatial resolution. To solve this problem, we urgently need a backbone network that can maintain a high resolution.

Recently, the HRFPN has achieved promising results for region-level ship detection in both inshore and offshore areas of SAR images [10]. The HRFPN invariably maintains HR feature maps by connecting parallel high-to-low resolution convolutions, and repeatedly exchange the information between multi-resolution representations. In addition, FPN is a serial connection, and HRFPN is a parallel connection. Hence, compared with FPN, the final feature maps are semantically richer and spatially more accurate. Nowadays, to fully utilize multi-level feature maps and maintain HR feature maps, we introduce the HRFPN into pixel-level instance segmentation in remote sensing imagery.

As in [10], the framework of the HRFPN consists of four stages of parallel convolution streams and an HRFPN block. A detailed description of the four-phase parallel convolutional flow can be found in the literature [10,40,41].

The detailed description of the HRFPN block is shown in Figure 4. Firstly, we represent the four outputs from high- to low-resolution as $\left\{C_2, C_3, C_4, C_5\right\}$. Then, the feature maps of all parallel convolutions are aggregated, and the result is defined as $P_2$. Finally, the feature maps with the same spatial size in the top-down pathway and $\left\{C_2, C_3, C_4, C_5\right\}$ are merged through lateral connections. Besides, to decrease the aliasing impact of sampling, a $3 \times 3$ convolutional layer is attached to each merged map to produce the final feature map. This final set of feature maps are defined as $\left\{P_2, P_3, P_4, P_5\right\}$, corresponding to $\left\{C_2, C_3, C_4, C_5\right\}$. Especially, the channel dimension in each feature map is reduced via a $1 \times 1$ convolutional layer. The output channels of HRFPN is set to 256. The entire process of the HRFPN block is as follows

$$P_2 = Conv_{1\times1}(C_2) \oplus Upsample(C_3) \oplus Upsample(C_4) \oplus Upsample(C_5)$$
$$P_{i+1} = Conv_{3\times3}\left[Conv_{1\times1}(C_{i+1}) \oplus Downsample(P_i)\right], i = 2,3,4 \tag{1}$$

where $Conv_{1\times1}$ and $Conv_{3\times3}$ indicate a $1\times1$ convolution layer and a $3\times3$ convolution layer, respectively; *Upsample* indicates bilinear up-sampling and then performs a $1\times1$ convolution; *Downsample* indicates a $3\times3$ convolution layer with a stride of 2, respectively; $\oplus$ indicates the operation of concatenation.

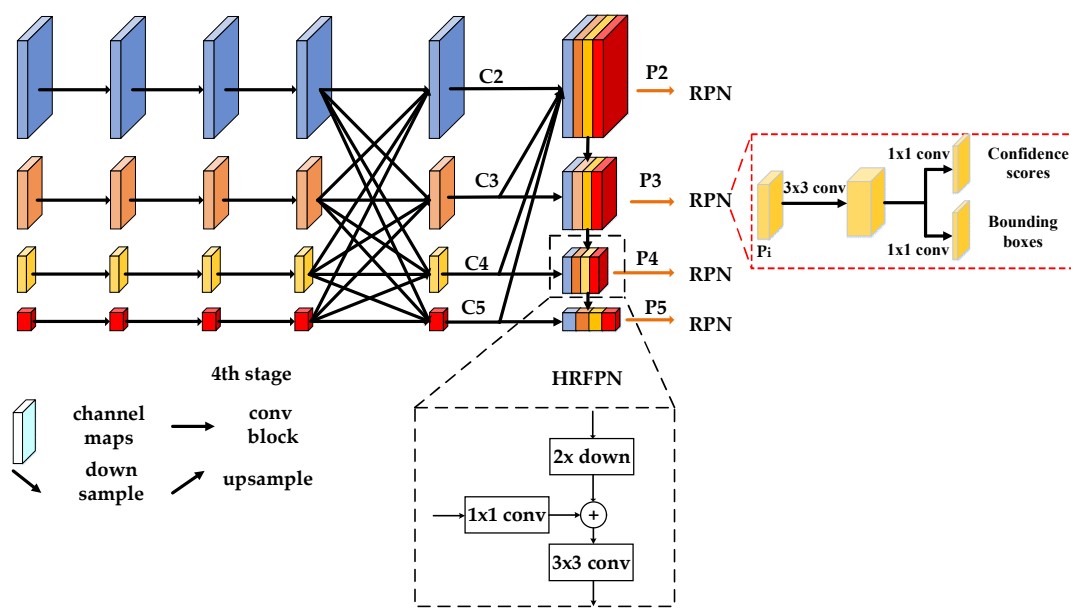

**Figure 4.** Illustration of the HR feature pyramid network (HRFPN) block.

Furthermore, the candidate proposals are generated by the region proposal network (RPN) [33,34]. Specifically, HRFPN's output $P_i$ generates candidate proposals through a $3\times3$ convolution and two sibling $1\times1$ convolutions, as shown on the right side of Figure 4. In RPN, anchors are often involved. Following the literature [10,33–35], the areas of the anchors are set to $\{32^2, 64^2, 128^2, 256^2, 512^2\}$ pixels on five stages $\{P_2, P_3, P_4, P_5, P_6\}$ respectively, where $P_6$ is obtained via a $3\times3$ convolutional layer with a stride of 2 on $P_5$. The anchors of multiple aspect ratios are used $\{1:2, 1:1, 2:1\}$ at each stage. Thus, there are a total of 15 anchors on the pyramid. For other descriptions of RPN, please refer to [10,34].

3.1.2. Instance Segmentation Network

Cai et al. [21] proposed a multi-stage architecture for object detection and instance segmentation called Cascade Mask R-CNN, which achieves promising results due to the adaptive handling of training distributions and progressive refinement of predictions. Therefore, we will implement our instance segmentation method based on Cascade Mask R-CNN to perform high-quality instance segmentation.

Cascade Mask R-CNN is obtained by direct hybridization of Cascade R-CNN and Mask R-CNN. In this implementation, each stage is similar to Mask R-CNN [19], with a mask branch, a class branch, and a box branch. The current stage will accept RPN or the box returned by the previous stage as an input, and then predict the new box and mask. For the convenience of

description, we refer to the instance segmentation part in Cascade Mask R-CNN as ISNetV1, as illustrated in Figure 5a.

In ISNetV1, RoIAlign [19] is used to extract regional features from the proposals generated by RPN or the bounding box regression of the previous stage. Specifically, all proposals are adjusted to $7 \times 7$ and $14 \times 14$ by RoIAlign for the box branch and mask branch, respectively [19,21]. As is shown in Figure 5, the intersection over the union (IoU) thresholds of three detection heads are 0.5, 0.6, and 0.7, in which the predictions of each stage are fed into the next stage to obtain high-quality prediction results. The detection heads in the ISNetV1 have the same architecture [21]. Besides, the box branches and the class branches are consistent with the literature [10,21]. The mask branch is a small fully convolutional network (FCN) applied to each Region of Interest (RoI), predicting an instance mask in a pixel-to-pixel manner. Moreover, the mask branch generates small feature maps $(28 \times 28)$ from each proposal through four $3 \times 3$ convolutional layers and one deconvolutional layer [19]. Finally, the ISNetV1 is executed to obtain the final instance segmentation results.

Although the ISNetV1 has achieved good results, the mask prediction performance can still be improved. As can be seen in Figure 5a, the three mask branches of ISNetV1 lack direct information flow. The instance mask prediction of each stage completely depends on the bounding box regression of the previous stage and the RoI features of the current stage, without any connection with the mask branch of the previous stage. Specifically, the mask branches of multiple stages are more like training with the data of different distributions and then the ensemble during testing, rather than playing the role of gradual adjustment and enhancement between stages, which will prevent further improvements in mask prediction precision.

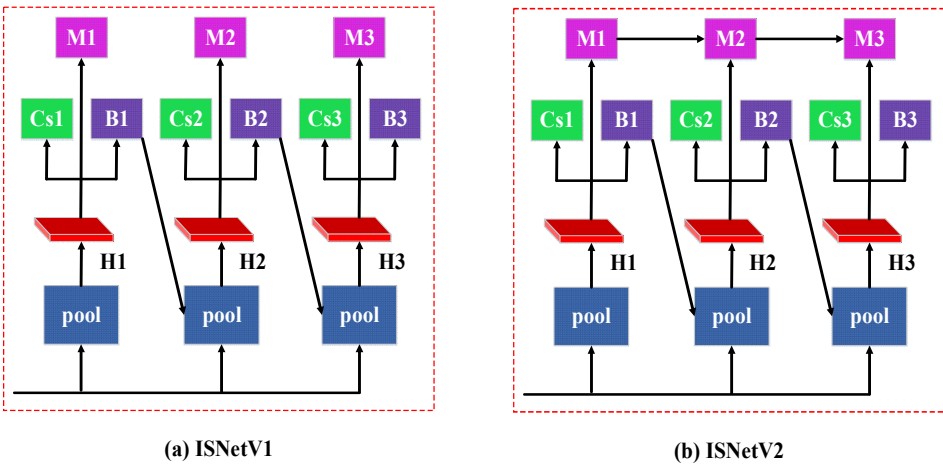

**Figure 5.** The architecture of the ISNet method where "Cs" denotes the classification; "B" indicates the bounding box; "M" represents the mask branch; "H" denotes the detection head; "pool" represents the region feature extraction. (**a**) ISNetV1; (**b**) ISNetV2.

In Cascade R-CNN [35], the information flow of the box branch is to make the features and learning goals of the next stage relevant to the current stage, that is, to gradually improve the prediction between different stages.

To address this problem, we follow similar principles in Cascade R-CNN [35], adding a connection between the mask branches of adjacent stages to provide the information flow of the mask branches, as illustrated in Figure 5b. Specifically, the mask features from the previous stage are provided to the current stage to facilitate further interaction of the information flow. The optimized network is called ISNetV2, as illustrated in Figure 5b.

In ISNetV2, the mask branch $M_i$ is a small FCN, which consists of four consecutives $3 \times 3$ convolutional layers and one deconvolutional layer, as shown in Figure 6. The features of $M_i$ are subjected to feature embedding through a $1 \times 1$ convolution, and then input to $M_{i+1}$. In other

words, the feature maps before the deconvolutional layer are then embedded with a $1\times1$ convolutional layer to align with the merged backbone features of RoI. Lastly, the result is added to the next RoI through the element-wise sum. The rest of ISNetV2 is consistent with ISNetV1. The ISNetV2 uses the introduced bridge to directly interact with the adjacent mask branches, instead of separating mask features, which will promote further improvements in mask prediction accuracy.

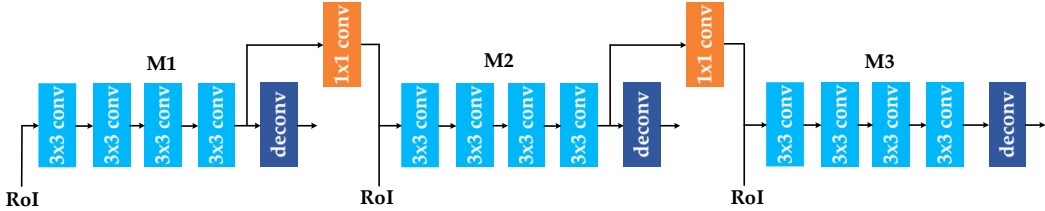

**Figure 6.** The architecture of three-stage mask branches.

*3.2. Loss Function*

For an image, during training, a multi-task loss function is as follows [19–21,32,33,35]

$$L = R_{cls} + R_{box} + R_{mask}. \tag{2}$$

where $R_{cls}$, $R_{box}$, and $R_{mask}$ represent the classification loss, the regression loss, and the segmentation loss, respectively.

The bounding box regression loss $R_{box}$ is defined as [21,35]

$$R_{box}\left[f\right] = \sum_i L_{box}\left(\mathbf{b}_i, \mathbf{g}_i\right). \tag{3}$$

where $\mathbf{g} = \left(g_x, g_y, g_w, g_h\right)$ and $\mathbf{b} = \left(b_x, b_y, b_w, b_h\right)$ can represent ground-truth bounding box and the predicted bounding box, respectively. As in [32,33],

$$L_{box}\left(\mathbf{b}, \mathbf{g}\right) = \sum_{j\in\{x,y,w,h\}} smooth_{L_1}\left(b_j - g_j\right). \tag{4}$$

in which

$$smooth_{L_1}\left(x\right) = \begin{cases} 0.5x^2, |x| < 1 \\ |x| - 0.5, otherwise \end{cases}. \tag{5}$$

is the smooth $L_1$ loss. $smooth_{L_1}$ operates on the distance vector $\Delta = \left(\delta_x, \delta_y, \delta_w, \delta_h\right)$ defined by [32,33,35]

$$\delta_x = \left(g_x - b_x\right)/b_w, \delta_y = \left(g_y - b_y\right)/b_h$$
$$\delta_w = \log\left(g_w / b_w\right), \delta_h = \log\left(g_h / b_h\right) \tag{6}$$

In addition, $\Delta = \left(\delta_x, \delta_y, \delta_w, \delta_h\right)$ needs to be normalized [32,33,35].

The classification loss $R_{cls}$ is defined as follows

$$R_{cls}\left[h\right] = \sum_i L_{cls}\left(p_i, y_i\right). \tag{7}$$

where

$$L_{cls}\left(p, y\right) = -\log p_y \tag{8}$$

is the cross-entropy loss. $y$ is the class label. $p$ is a discrete probability distribution over the $M+1$ categories.

The mask branch has a $K \times m \times m$ dimensional output for each RoI, which encodes $K$ binary masks of resolution $m \times m$, one for each of the $K$ classes. The segmentation risk can be minimized as follows

$$R_{mask} = \sum_i L_{mask}\left(m_i^s, \hat{m}_i^s\right).$$  (9)

where $L_{mask}$ is the binary cross-entropy loss form in Mask R-CNN [19]. $m_i^s$ and $\hat{m}_i^s$ represent the mask predictions and ground-truth mask, respectively.

## 4. Experiments

In this section, the instance segmentation approaches will be evaluated in high-resolution remote sensing imagery.

### 4.1. Dataset Description

Two datasets are used in our experiments, including the SSDD dataset and the NWPU VHR-10 dataset. Instance masks in SSDD dataset and NWPU VHR-10 dataset have been released in https://github.com/chaozhong2010/VHR-10_dataset_coco.

#### 4.1.1. The SSDD Dataset

The SSDD datasets [23] include 1160 SAR images with resolutions ranging from 1 to 15 m. Besides this, the SSDD has a total of 2540 ships. We further mark the instance masks directly on the SSDD dataset. In this paper, we use the LabelMe [42] open source project on GitHub to annotate these SAR images. Then, LabelMe converts the annotation message into the COCO JSON format. The SAR images annotation process is shown in Figure 7. In all experiment, the datasets are randomly split into a train dataset 70% (812 images) and a test dataset 30% (348 images).

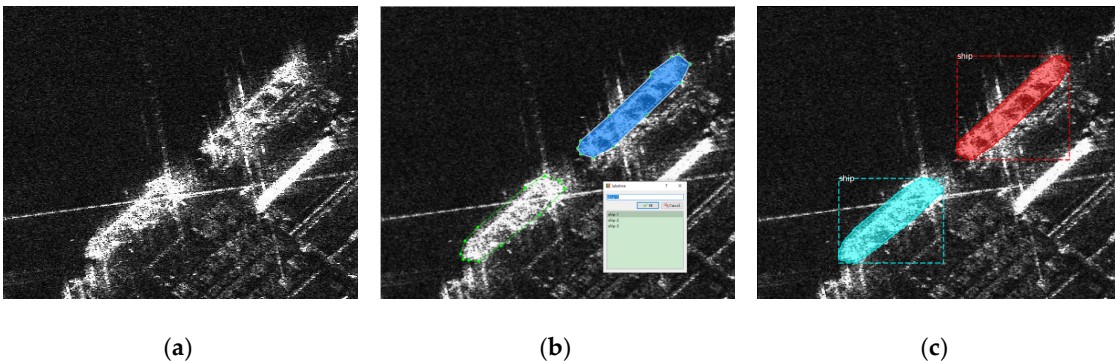

(**a**) (**b**) (**c**)

**Figure 7.** Example images and annotated instance masks of the synthetic aperture radar ship detection dataset (SSDD). (**a**) original synthetic aperture radar (SAR) images; (**b**) labeling process; (**c**) visualization result.

#### 4.1.2. The NWPU VHR-10 Dataset

The experiment also uses the NWPU VHR-10 datasets [3,6], which is a challenging ten-class geospatial object detection dataset. The positive image set in the datasets contains a total of 650 high-resolution optical remote sensing images with a resolution ranging from 0.08 to 2 m. These images were acquired from Google Earth and Vaihingen data. Su et al. [3] manually used the instance masks to annotate ten-class objects in these optical remote sensing images. Then, LabelMe converts the annotation message into the COCO JSON format. Some examples of images and the corresponding annotated instance masks are shown in Figure 8. In all experiment, the datasets are randomly split into a train dataset 70% (455 images) and a test dataset 30% (195 images).

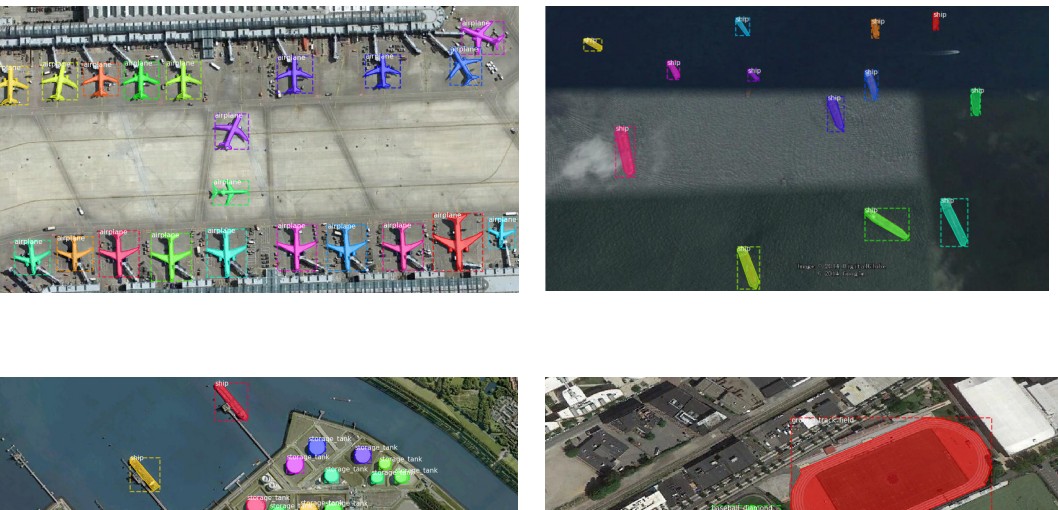

**Figure 8.** Example images and annotated instance masks of the NWPU VHR-10 data set.

## 4.2. Evaluation Metrics

For the instance segmentation of remote sensing imagery, the intersection over union (IoU) is the overlap rate between the ground-truth and the predicted mask. The calculation formulas of IoU is as follows

$$IoU\left(M_p, M_g\right) = \frac{M_p \cap M_g}{M_p \cup M_g}.$$

(10)

where $M_p$ represents the predicted mask and $M_g$ denotes the ground-truth mask.

The performance of the instance segmentation methods in remote sensing images is quantitatively and comprehensively evaluated by the standard COCO [43] metrics. These metrics include average precision (AP), $AP_{50}$, $AP_{75}$, $AP_S$, $AP_M$, $AP_L$ [43]. The average precision (AP) is averaged across all 10 IoU thresholds (0.50: 0.05: 0.95) and all categories. Averaging over IoUs rewards detectors with better localization. The larger AP value indicates that the more accurate the predicted instance masks, the better the instance segmentation performance. $AP_{50}$ represents the calculation under the IoU threshold of 0.50; $AP_{75}$ is a stricter metric and represents the calculation under the IoU threshold of 0.75. Therefore, $AP_{75}$ performs better than $AP_{50}$ in the instance of mask accuracy evaluation. The greater $AP_{75}$ value indicates more accurate instance masks. $AP_L$ is set for large targets (area > $96^2$); $AP_M$ is set for medium targets ($32^2$ < area < $96^2$); $AP_S$ is set for small targets (area < $32^2$).

## 4.3. Implementation Details

All the experiments are implemented on pytorch and mmdetection [44]. The operating system is Ubuntu 16.04. A single GTX-1080Ti GPU is used to train and test the detectors.

In our experiments, the HQ-ISNet uses HRFPN-W18, HRFPN-W32, and HRFPN-W40 for feature extraction, in which 18, 32, and 40 indicate the widths of the HR subnetworks, respectively. Regarding HRFPN-W18, HRFPN-W32, and HRFPN-W40, the dimensions of the HR representation are reduced to 144, 256, and 320, respectively, by a $1 \times 1$ convolution. For HRFPN-W18, the output channels of the four-resolution feature maps are 18, 36, 72, and 144. For HRFPN- W32, they are 32,

64, 128, and 256. For HRFPN-W40, they are 40, 80, 160, and 320. In addition, the IoU thresholds of the three detection heads were set to 0.5, 0.6, and 0.7, respectively.

The comparative experiments are performed using advanced object detection and instance segmentation approaches: Faster R-CNN [33], Mask R-CNN [19], Cascade R-CNN [35], Cascade Mask R-CNN [20,21], Mask Scoring R-CNN [22], and Hybrid Task Cascade [20]. They all use ResNet-FPN [45,46] as backbone networks.

For HQ-ISNet, Hybrid Task Cascade, and Cascade Mask R-CNN, we use a single GPU to train the model for 20 epochs [20,21,41,44]. The initial learning rate (LR) is set as 0.0025 for these methods. Then, the LR will gradually reduce by 0.1 after 16 and 19 epochs, respectively. The batch size is set to two images. We train Faster R-CNN, Mask R-CNN, Cascade R-CNN, and Mask Scoring R-CNN with batch size of 2 for 12 epochs [20,21,41,44]. The initial learning rate is set as 0.0025 for these methods. Then, the learning rate will gradually reduce by 0.1 after eight and 11 epochs, respectively. Besides, SGD is used to optimize the entire model. We use a momentum of 0.9 and a weight decay of 0.0001. The input images are adjusted to 1000 px along the long axis and 600 px along the short axis by the bilinear interpolation. Additionally, the overall framework is optimized in an end-to-end manner. All other hyper-parameters follow the literature [10,19–22,33,35,44] in this paper.

*4.4. Results and Analysis of HQ-ISNet*

4.4.1. Results of the HQ-ISNet

The instance segmentation outcomes of the proposed approach in SAR images and remote sensing optical images are shown in Figure 9. a and c are ground-truth mask; b and d are the predicted instance outcomes. As can be seen in Figure 9, HQ-ISNet is suitable for our instance segmentation task in HR remote sensing images. HQ-ISNet has almost no missed detections and false alarms, which guarantees that our mask branch performs instance segmentation. Finally, these artificial targets are correctly detected and segmented. Moreover, the segmentation results of HQ-ISNet are very close to the ground truth. HQ-ISNet successfully completed the instance segmentation task in HR remote sensing images.

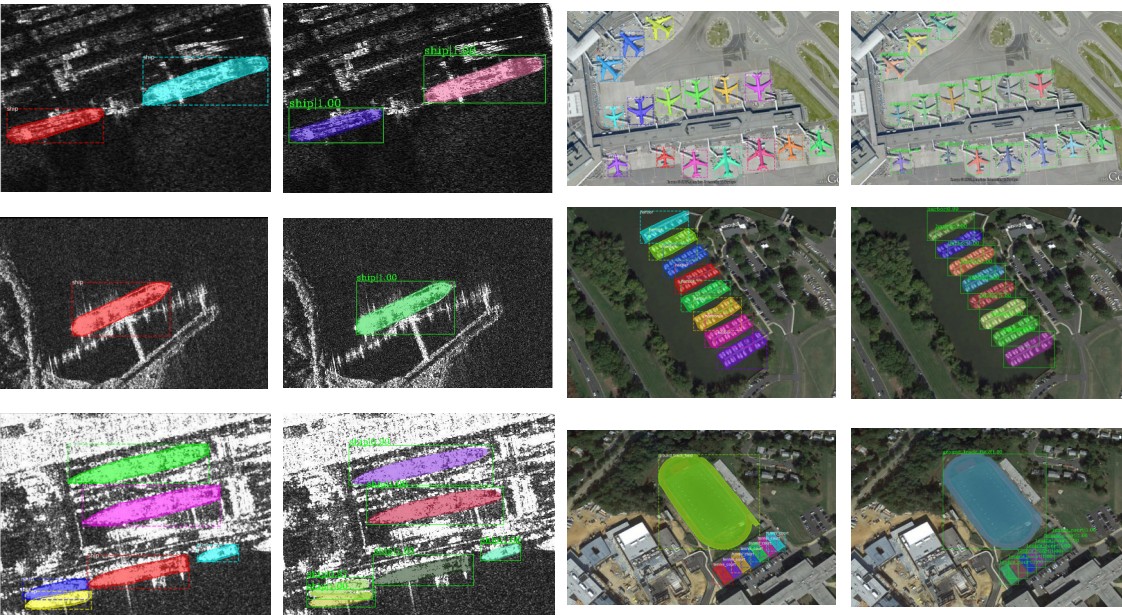

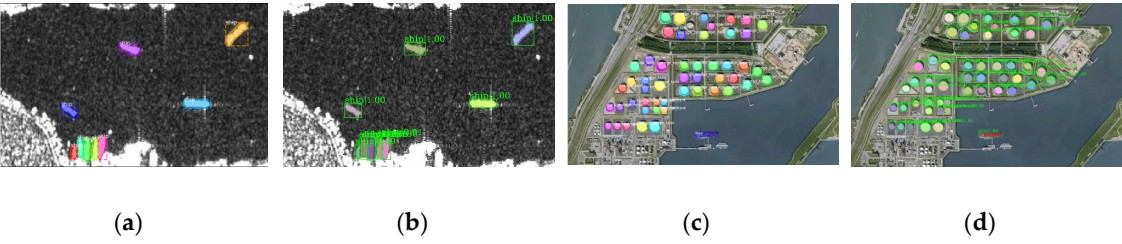

(**a**)          (**b**)          (**c**)          (**d**)

**Figure 9.** Instance segmentation outcomes of the proposed approach in SAR images and remote sensing optical images. (**a**) and (**c**) are ground-truth mask; (**b**) and (**d**) are predicted instance results.

To further test our network, we performed test experiments on the SAR image from the port of Houston. SAR images were obtained with a Sentinel-1B [47] sensor. The following is the parameter information: the resolution is 3m, the polarization method is HH, and the imaging mode is S3-StripMap. In addition, we have annotated according to the labeling methods and principles in Section 4.1.1. As can be seen from Figure 10, HQ-ISNet successfully completed the instance segmentation task in the SAR image. Our results have almost no missed ships and false alarms, and the segmentation results also are very close to the ground truth.

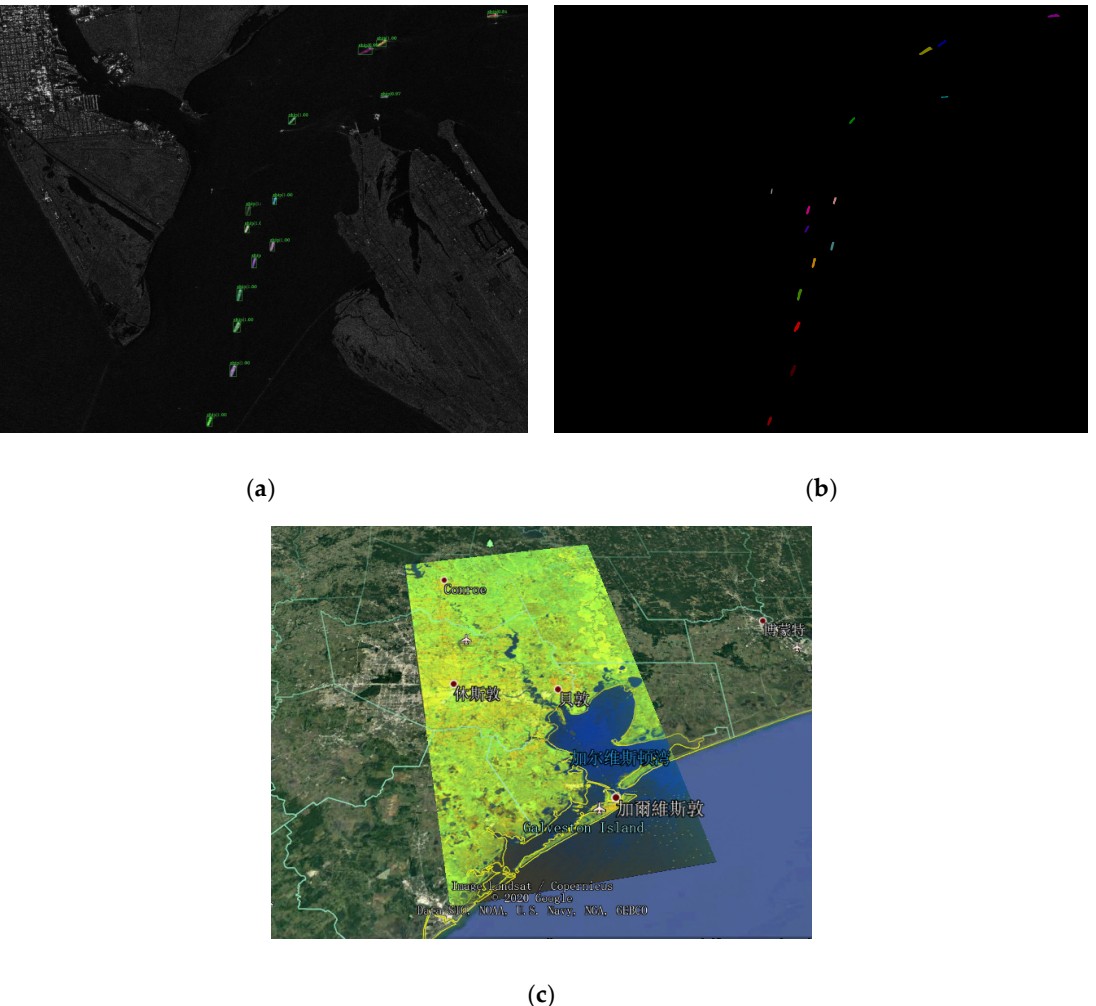

**Figure 10.** Instance segmentation outcomes with the HQ-ISNet on the SAR image from the port of Houston. (**a**) Final results; (**b**) mask instance labels; (**c**) Sentinel-1B sensor imaging area.

From Table 1 and Table 2, we can see that the HQ-ISNet, based on the ISNetV2 module and HRFPN backbone, has the best instance segmentation performance. It achieves 67.4% and 67.2% AP on the SSDD dataset and NWPU VHR-10 dataset, respectively. More specifically, with the help of the HRFPN, our network achieves a 2.1% and 5.3% performance improvement on the SSDD dataset and NWPU VHR-10 dataset in terms of AP. With the help of the ISNetV2, our network achieves 1.3% and 4.8% performance improvement on the SSDD dataset and NWPU VHR-10 dataset in terms of AP. Moreover, for $AP_{75}$ score, our network achieves a gain of 1% and 1.6% on the SSDD dataset with ISNetV2 and HRFPN, respectively. For the $AP_{75}$ value, our network achieves a gain of 5.4% and 6.1% on the NWPU VHR-10 dataset with ISNetV2 and HRFPN, respectively. In the SSDD and NWPU VHR-10 dataset, the $AP_{50}$ has also been increased. Besides this, $AP_S$ value, $AP_M$ value, and $AP_L$ value have also been improved in the SSDD and NWPU VHR-10 dataset. Among them, the $AP_L$ in SSDD has a decline rate under the influence of HRFPN. We will discuss this in Section 5. The HRFPN can maintain a high resolution and solve the problem of spatial resolution loss in FPN, and the ISNetV2 refines the mask information flow between mask branches. Accordingly, the final feature maps are semantically richer and spatially more accurate. The final predicted instance mask is also more accurate. In addition, there are only ships in SSDD, and ten target categories in NWPU VHR-10 involve ship, harbor, ground track field, basketball court, etc. The NWPU VHR-10 dataset with many complex targets needs this rich semantic and spatial information, so its improvement is the most obvious. The results reveal that the HRFPN and ISNetV2 modules can effectively improve instance segmentation performance in remote sensing images.

**Table 1.** Results on SSDD for HQ-ISNet. The first line is the original baseline results; the second line is the result of HRFPN replacing FPN; the third line is the result by adding ISNetV2; the fourth line is the result obtained by the combined operation of HRFPN and ISNetV2.

| FPN | HRFPN | ISNetV1 | ISNetV2 | AP | $AP_{50}$ | $AP_{75}$ | $AP_S$ | $AP_M$ | $AP_L$ |
|---|---|---|---|---|---|---|---|---|---|
| ✓ | | ✓ | | 65.1 | 94.8 | 83.4 | 65.7 | 65.0 | 20.0 |
| | ✓ | ✓ | | 67.2 | 95.6 | 85.0 | 66.7 | 68.9 | 16.7 |
| ✓ | | | ✓ | 66.4 | 96.1 | 84.4 | 66.3 | 67.7 | 53.6 |
| | ✓ | | ✓ | 67.4 | 96.4 | 85.8 | 67.2 | 69.5 | 54.5 |

**Table 2.** Results on NWPU VHR-10 for HQ-ISNet. The first line is the original baseline results; the second line is the result of HRFPN replacing FPN; the third line is the result by adding ISNetV2; the fourth line is the result obtained by the combined operation of HRFPN and ISNetV2.

| FPN | HRFPN | ISNetV1 | ISNetV2 | AP | $AP_{50}$ | $AP_{75}$ | $AP_S$ | $AP_M$ | $AP_L$ |
|---|---|---|---|---|---|---|---|---|---|
| ✓ | | ✓ | | 60.3 | 92.3 | 66.6 | 45.3 | 60.7 | 67.3 |
| | ✓ | ✓ | | 65.6 | 94.5 | 72.7 | 52.7 | 66.0 | 77.9 |
| ✓ | | | ✓ | 65.1 | 94.5 | 72.0 | 49.6 | 65.9 | 76.6 |
| | ✓ | | ✓ | 67.2 | 94.6 | 74.2 | 52.1 | 67.8 | 77.5 |

### 4.4.2. Effect of HRFPN

The comparison of the outcomes of HRFPN and FPN in SAR images and remote sensing optical images is displayed in Figure 11. Mask R-CNN is used as a powerful baseline to accomplish our approach and comparison approach. Compared with FPN, the segmentation results of HRFPN are closer to the ground truth mask, and the instance masks of HRFPN are more accurate. It is worth noting that the instance segmentation performance of the HRPFN is better than the original FPN for the high-resolution remote sensing imagery.

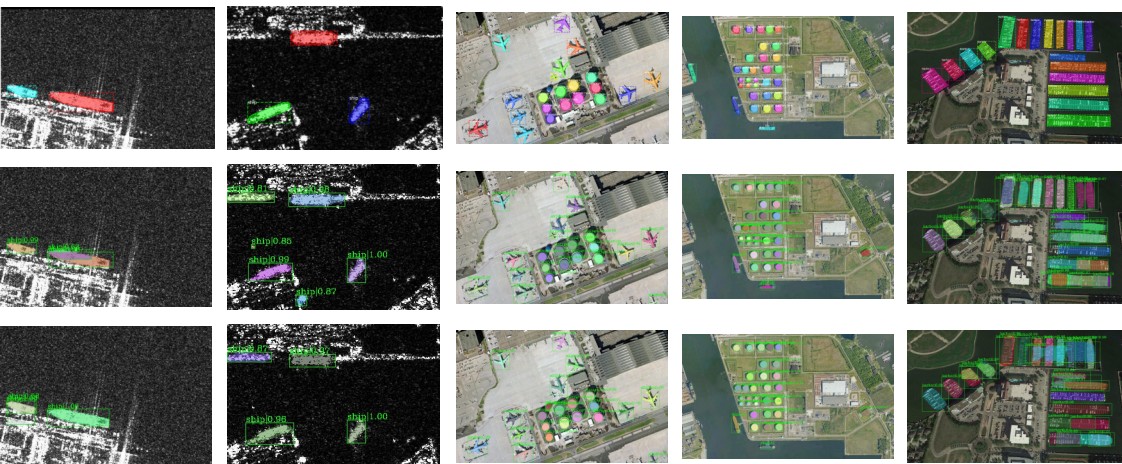

**Figure 11.** Outcomes of FPN and HRFPN in SAR images and remote sensing optical images. Row 1 is ground-truth mask; Row 2 and Row 3 are the outcomes of FPN and HRFPN, respectively.

From Table 3 and Table 4, we can see that the HRFPN is more efficient than FPN in the Mask R-CNN framework for instance segmentation, with less computational complexity and smaller parameters. The AP value is 66.0% on the SSDD dataset, which can achieve a performance improvement of nearly 1.5% compared to FPN. In addition, the AP value is 60.7% on the NWPU VHR-10 dataset, which can achieve a performance improvement of nearly 3.3% compared to FPN. It has been suggested that our approach acquires more accurate instance masks and improves the instance segmentation performance. The $AP_{50}$ and $AP_{75}$ scores are 96.2% and 85.0% on the SSDD dataset, which achieves a 0.5% and 2.4% performance improvement over the FPN, respectively. Moreover, the $AP_{50}$ and $AP_{75}$ scores are 92.7% and 65.5% on the NWPU VHR-10 dataset, which achieves a 1% and 2.7% performance improvement over the FPN, respectively. We find that $AP_{75}$ improves significantly compared to $AP_{50}$ on both datasets. With looser metrics, $AP_{50}$, our method may approach the best performance in the two datasets, so the improvement is not significant. However, under more stringent indicators $AP_{75}$, our method has been greatly improved. Therefore, the predicted instance masks are more accurate. Besides, the performance improvement is obtained for small ships ($AP_S$) on the SSDD dataset, and $AP_M$ maintains the original performance. We disucss $AP_L$ in Section 5. Importantly, the $AP_S$ score, $AP_M$ score, and $AP_L$ score have been greatly increased on the NWPU VHR-10 dataset. In particular, the performance improvement of small targets is most obvious, and small targets achieve nearly 6.2% performance gains.

Furthermore, we find that HRFPN improves the NWPU VHR-10 dataset more significantly than the SSDD dataset. There are only ships in SSDD, and ten target categories in NWPU VHR-10 involve ship, harbor, ground track field, basketball court, etc. Our HRFPN can maintain a high resolution and solve the problem of spatial resolution loss in FPN. Hence, the final feature maps are semantically richer and spatially more accurate compared with FPN. The NWPU VHR-10 dataset with many complex targets needs this rich semantic and spatial information, so its improvement is the most obvious, especially for small targets. In short, HRFPN can effectively improve instance segmentation performance in remote sensing images, with less computational complexity and smaller parameters.

**Table 3.** Influence of the HRFPN on the SSDD dataset. Where "R-50" indicates ResNet-50; "R-101" represents ResNet-101.

| Backbone | AP | $AP_{50}$ | $AP_{75}$ | $AP_S$ | $AP_M$ | $AP_L$ | Time (ms) | Param (M) | Flops |
|---|---|---|---|---|---|---|---|---|---|
| R-50-FPN | 64.4 | 95.1 | 81.0 | 65.4 | 62.3 | 12.7 | 51.8 | 43.75 | 198.02 |
| R-101-FPN | 64.5 | 95.7 | 82.6 | 64.7 | 65.0 | 22.0 | 63.3 | 62.74 | 244.27 |
| HRFPN-W18 | 65.0 | 95.7 | 82.7 | 65.8 | 63.5 | 13.4 | 65.8 | 29.71 | 186.13 |
| HRFPN-W32 | 65.5 | 95.8 | 84.0 | 66.2 | 65.3 | 20.6 | 74.1 | 49.50 | 245.65 |

| HRFPN-W40 | 66.0 | 96.2 | 85.0 | 66.5 | 65.5 | 15.1 | 86.2 | 65.75 | 293.56 |

In the HRFPN structure, the HRFPN-W40 achieves a 66.0% AP score on the SSDD dataset and a 60.7% AP score on the NWPU VHR-10 dataset, which is improved compared with HRFPN-W18 and HRFPN-W32, but it also increases computational complexity and the parameters.

In conclusion, the HRFPN, which fully utilizes multi-level feature maps and can maintain HR feature maps, can make the predicted instance masks more accurate and effectively improve the instance segmentation performance for the HR remote sensing images.

**Table 4.** Influence of the HRFPN on the NWPU VHR-10 dataset. Where "R-50" indicates ResNet-50; "R-101" represents ResNet-101.

| Backbone | AP | AP$_{50}$ | AP$_{75}$ | AP$_S$ | AP$_M$ | AP$_L$ | Time (ms) | Param (M) | Flops |
|---|---|---|---|---|---|---|---|---|---|
| R-50+FPN | 56.2 | 90.2 | 60.7 | 40.9 | 56.6 | 61.1 | 61.0 | 43.80 | 198.25 |
| R-101-FPN | 57.4 | 91.7 | 62.8 | 41.0 | 57.5 | 60.5 | 71.4 | 62.79 | 244.5 |
| HRFPN-W18 | 58.0 | 89.9 | 64.9 | 43.3 | 58.9 | 64.3 | 75.2 | 29.75 | 186.36 |
| HRFPN-W32 | 59.7 | 91.1 | 64.7 | 46.3 | 60.1 | 64.0 | 83.3 | 49.55 | 245.87 |
| HRFPN-W40 | 60.7 | 92.7 | 65.5 | 47.2 | 61.6 | 64.0 | 96.2 | 65.80 | 293.79 |

### 4.4.3. Effect of ISNetV2

The comparison results of ISNetV1 and ISNetV2 in SAR images and remote sensing optical images are displayed in Figure 12. Cascade Mask R-CNN is used as a powerful baseline to accomplish our approach and comparison approach. Compared with ISNetV1, the segmentation result of ISNetV2 is closer to the ground truth mask. The ISNetV2 is more accurate than ISNetV1 in the mask segmentation. There is no doubt that that the instance segmentation performance of the ISNetV2 is better than the original ISNetV1 for the high-resolution remote sensing imagery, especially for high-resolution artificial targets.

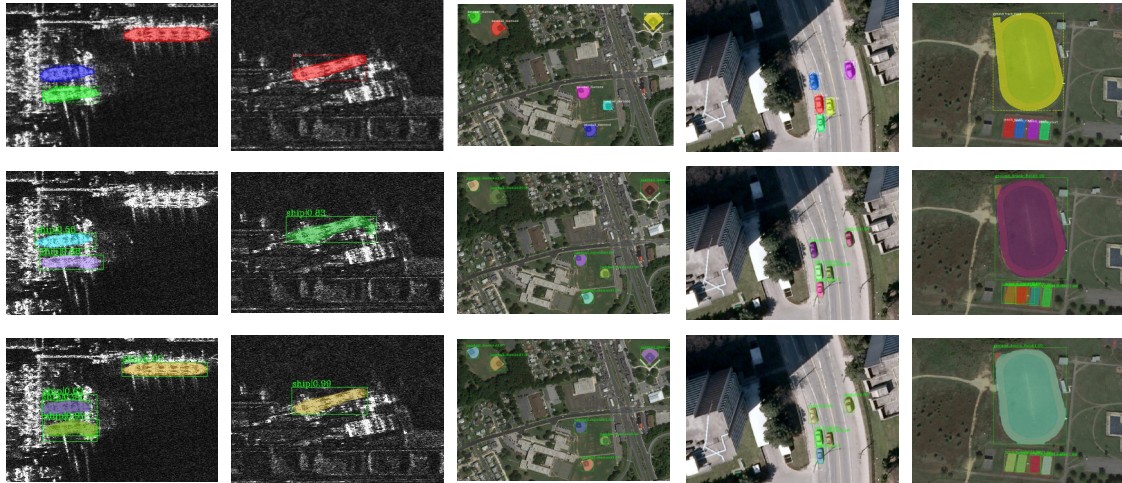

**Figure 12.** Comparison results of ISNetV1 and ISNetV2 in SAR images and remote sensing optical images. Row 1 is ground-truth mask; Row 2 is the outcome of ISNetV1; Row 3 is the outcome of ISNetV2.

From Table 5 and Table 6, we can see that the ISNetV2 is more efficient than ISNetV1 in the Cascade Mask R-CNN framework for instance segmentation, with similar parameters and computational cost. With the ISNetV2, Cascade Mask R-CNN performs better on the SSDD dataset, which can achieve a performance improvement of nearly 1.4% in terms of AP. In addition, the AP value on the NWPU VHR-10 dataset achieves nearly 4.8% performance gains over ISNetV1. It has been suggested that ISNetV2 refines the mask information flow between the mask branches, which

promotes further improvements in mask prediction accuracy. The $AP_{50}$ and $AP_{75}$ scores on the SSDD dataset, compared to ISNetV1, achieve a gain of 1.3% and 1%, respectively. Moreover, the $AP_{50}$ and $AP_{75}$ scores on the NWPU VHR-10 dataset, compared to ISNetV1, achieve a gain of 2.2% and 5.8%, respectively. The results reveal that the predicted instance mask is more accurate. Importantly, a large performance improvement is obtained for medium ships ($AP_M$) and large ships ($AP_L$) on the SSDD dataset, and $AP_S$ also improved. Besides these, the $AP_S$ score, $AP_M$ score, and $AP_L$ score have been greatly increased on the NWPU VHR-10 dataset. As a result, the instance segmentation performance is remarkably enhanced for small, medium and large targets. Furthermore, we find that ISNetV2 improves the NWPU VHR-10 dataset more significantly than SSDD dataset. There are only ships in SSDD, and ten target categories in NWPU VHR-10 involve ship, harbor, ground track field, basketball court, etc. When calculating the IoU, we know that the larger the target, the more pixels it takes, and the inaccurate prediction has a great impact on the quantitative result. Therefore, small changes have a big impact on results. Our ISNetV2 improves the mask information flow and makes the predicted results more accurate. Therefore, the NWPU VHR-10 dataset with a larger target size has the most significant improvement.

**Table 5.** Influence of the ISNetV2 on the SSDD dataset. Where "R-50" indicates ResNet-50; "R-101" represents ResNet-101. "✓" means use ISNetV2 and "-" means use ISNetV1.

| Backbone | ISNetV2 | AP | $AP_{50}$ | $AP_{75}$ | $AP_S$ | $AP_M$ | $AP_L$ | Time (ms) | Param (M) | Flops |
|---|---|---|---|---|---|---|---|---|---|---|
| R-50-FPN | - | 65.1 | 94.8 | 82.6 | 65.7 | 64.4 | 20.0 | 72.5 | 76.80 | 359.65 |
| | ✓ | 65.9 | 96.1 | 83.5 | 66.0 | 66.9 | 30.8 | 71.9 | 76.99 | 362.24 |
| R-101-FPN | - | 65.0 | 94.8 | 83.4 | 65.5 | 65.0 | 12.0 | 86.2 | 95.79 | 405.90 |
| | ✓ | 66.4 | 95.8 | 84.4 | 66.3 | 67.7 | 53.6 | 84.7 | 95.99 | 408.49 |

In summary, the information flow between the mask branches is refined, which promotes further improvements in mask prediction accuracy. Therefore, ISNetV2 can effectively improve instance segmentation performance in the HR remote sensing imagery.

**Table 6.** Influence of the ISNetV2 on the NWPU VHR-10 dataset. Where "R-50" indicates ResNet-50; "R-101" represents ResNet-101. "✓" means use ISNetV2 and "-" means use ISNetV1.

| Backbone | ISNetV2 | AP | $AP_{50}$ | $AP_{75}$ | $AP_S$ | $AP_M$ | $AP_L$ | Time (ms) | Param (M) | Flops |
|---|---|---|---|---|---|---|---|---|---|---|
| R-50-FPN | - | 59.8 | 91.9 | 66.6 | 45.3 | 60.0 | 67.3 | 81.3 | 76.83 | 360.22 |
| | ✓ | 64.2 | 93.9 | 72.0 | 49.2 | 64.7 | 69.3 | 104.1 | 77.03 | 362.81 |
| R-101-FPN | - | 60.3 | 92.3 | 65.6 | 44.6 | 60.7 | 62.4 | 100.0 | 95.82 | 406.47 |
| | ✓ | 65.1 | 94.5 | 71.4 | 49.6 | 65.9 | 76.6 | 117.6 | 96.02 | 409.06 |

*4.5. Comparison with other approaches*

The qualitative outcomes between the HQ-ISNet and the comparison method on the SSDD dataset and NWPU VHR-10 dataset are displayed in Figures 13 and 14 to further validate the instance segmentation performance. Row 1 is ground-truth mask; Row 2-4 are the outcomes of Faster R-CNN, Cascade R-CNN, and Mask R-CNN, respectively; Row 5-7 are the outcomes of Cascade Mask R-CNN, Hybrid Task Cascade (HTC), and HQ-ISNet, respectively.

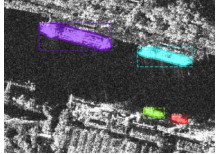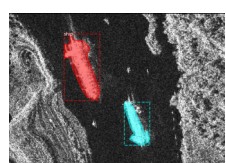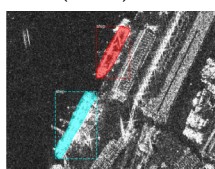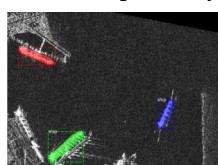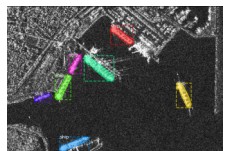

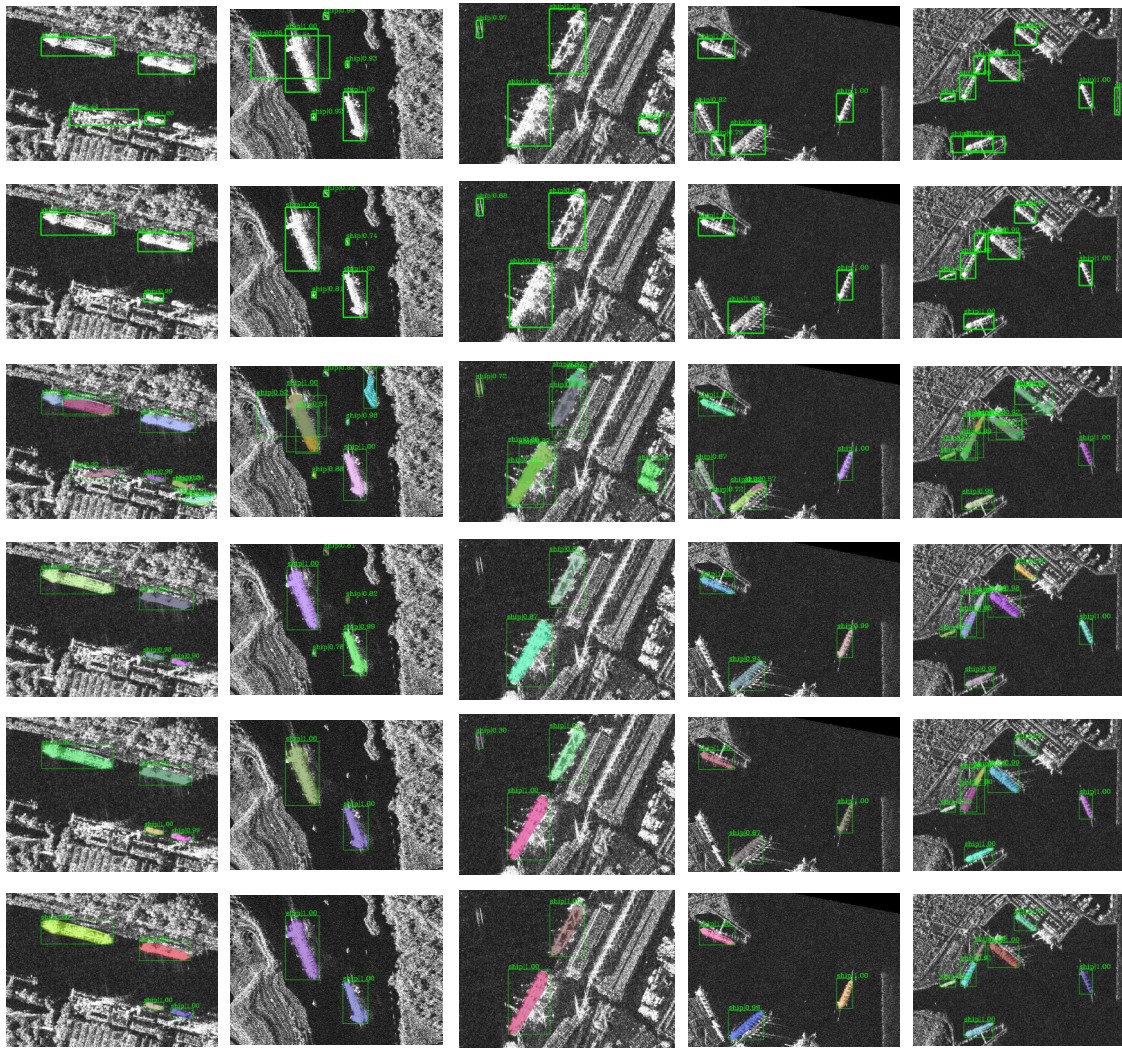

**Figure 13.** Outcomes from six approaches on the SSDD dataset. Row 1 is ground-truth mask; Row 2-4 are the outcomes of Faster R-CNN, Cascade R-CNN, and Mask R-CNN, respectively; Row 5-7 are the outcomes of Cascade Mask R-CNN, Hybrid Task Cascade, and HQ-ISNet, respectively.

As shown in Figures 13 and 14, compared with other instance segmentation methods, our approach can accurately detect and segment artificial targets in multiple remote sensing scenes. Specifically, these artificial targets are accurately covered by the predicted instance masks. HQ-ISNet has almost no missed detections and false alarms, which ensures that our mask branch performs better instance segmentation. Compared with bounding box detection, such as Faster R-CNN and Cascade R-CNN, the results of instance segmentation are closer to the silhouette of the original targets. The instance segmentation can also distinguish between different instances in the same category. The ships in Figure 13 are distinguished by different colors. The targets, such as airplanes, in Figure 14 are also distinguished by different colors. Furthermore, compared with other instance segmentation methods, our approach not only has almost no missed targets and false alarms but also has better mask segmentation results. The results of the SSDD dataset and the NWPU VHR-10 dataset imply that our method is suitable for instance segmentation task in HR remote sensing images and has a better mask segmentation performance than the other instance segmentation algorithms.

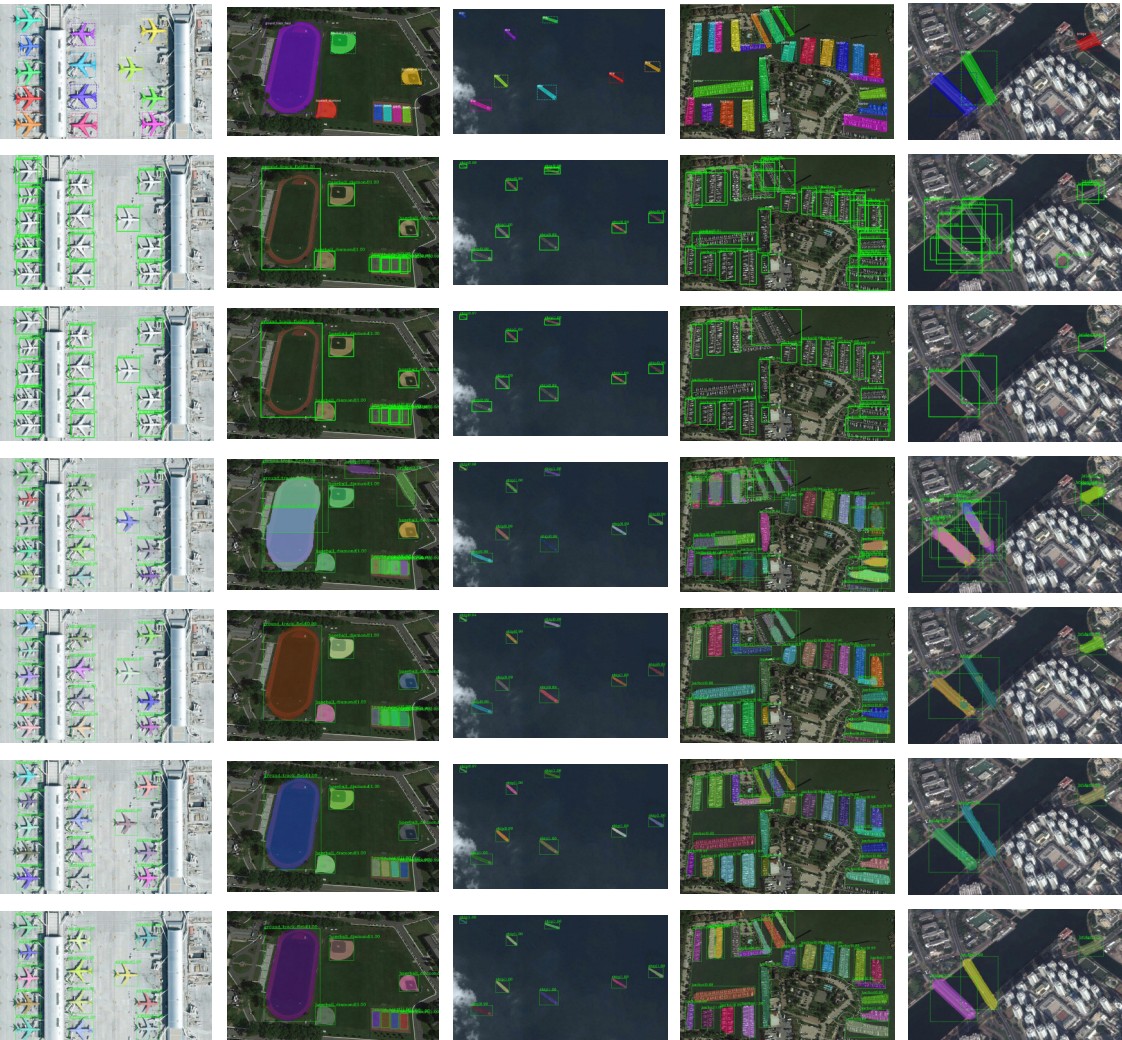

**Figure 14.** Outcomes from six approaches on the NWPU VHR-10 dataset. Row 1 is ground-truth mask; Row 2-4 are the outcomes of Faster R-CNN, Cascade R-CNN, and Mask R-CNN, respectively; Row 5-7 are the outcomes of Cascade Mask R-CNN, Hybrid Task Cascade, and HQ-ISNet, respectively.

In Tables 7 and 8, we compare the HQ-ISNet based on ISNetV2 and HRFPN with other advanced approaches on the SSDD dataset and the NWPU VHR-10 dataset to quantitatively evaluate the instance segmentation performance. These methods include Mask R-CNN [19], Mask Scoring R-CNN [22], Cascade Mask R-CNN [21] and Hybrid Task Cascade (HTC) [20] based on ResNet-FPN [45,46].

**Table 7.** Comparison to other advanced approaches on the SSDD dataset. Where "R-50" indicates ResNet-50 and "R-101" represents ResNet-101.

| Model | Backbone | Time (ms) | Model size | AP | $AP_{50}$ | $AP_{75}$ | $AP_S$ | $AP_M$ | $AP_L$ |
|---|---|---|---|---|---|---|---|---|---|
| Mask R-CNN | R-50-FPN | 51.8 | 351M | 64.4 | 95.1 | 81.0 | 65.4 | 62.3 | 8.5 |
| | R-101-FPN | 63.3 | 503M | 64.5 | 95.7 | 82.6 | 64.7 | 65.0 | 22.0 |
| Mask Scoring R-CNN | R-50-FPN | 50.8 | 481M | 64.1 | 94.2 | 81.0 | 64.8 | 62.8 | 11.7 |
| | R-101-FPN | 63.3 | 633M | 64.8 | 95.0 | 82.4 | 65.0 | 64.7 | 13.4 |
| Cascade Mask R-CNN | R-50-FPN | 72.5 | 615M | 65.1 | 94.8 | 82.6 | 65.7 | 64.4 | 20.0 |
| | R-101-FPN | 86.2 | 768M | 65.0 | 94.8 | 83.4 | 65.5 | 65.0 | 12.0 |

| | | | | | | | | | |
|---|---|---|---|---|---|---|---|---|---|
| Hybrid Task Cascade | R-50-FPN | 119.0 | 639M | 66.0 | 95.2 | 84.0 | 66.3 | 66.4 | 25.8 |
| | R-101-FPN | 133.3 | 791M | 66.9 | 95.3 | 85.0 | 66.6 | 67.7 | 29.6 |
| HQ-ISNet | HRFPN-W18 | 87.0 | 504M | 66.5 | 96.2 | 84.4 | 66.5 | 67.7 | 29.7 |
| | HRFPN-W32 | 93.5 | 662M | 67.3 | 96.3 | **85.8** | **67.2** | 68.8 | 24.1 |
| | HRFPN-W40 | 106.4 | 792M | **67.4** | **96.4** | 84.5 | 66.9 | **69.5** | **54.5** |

As can be observed from Table 7, the HQ-ISNet achieves the highest AP of 67.4%. Compared with Mask R-CNN and Mask Scoring R-CNN, the HQ-ISNet achieves 2.9% and 2.6% improvements, respectively. Besides, the HQ-ISNet achieves gains of 2.3% over Cascade Mask R-CNN. In short, compared with other instance segmentation algorithms on the SSDD dataset, our approach has a better instance segmentation performance and more accurate predicted instance masks. Moreover, the $AP_{50}$ score of HQ-ISNet is 96.4%, which has 0.7% improvements over Mask R-CNN, 1.4% gains over Mask Scoring R-CNN, and 1.6% improvements over Cascade Mask R-CNN. The HQ-ISNet attains an 85.8% $AP_{75}$, which achieves a gain of 3.2% over Mask R-CNN, 3.4% over Mask Scoring R-CNN, and 2.4% over Cascade Mask R-CNN. It has been established that the mask segmentation will be better and more precise than the other advanced approaches for instance segmentation on the SSDD dataset. The performance of small, medium, and large targets has also been improved on the SSDD dataset according to $AP_S$, $AP_M$, and $AP_L$. Under various AP indicators, we can obtain the same performance as HTC on the SSDD dataset, and some indicators exceed it, such as AP.

As can be observed from Table 8, the HQ-ISNet attains a 67.2% AP, which achieves a gain of 9.8% over Mask R-CNN, 8.4% over Mask Scoring R-CNN, and 6.9% over Cascade Mask R-CNN. In short, contrasted with other instance segmentation methods on the NWPU VHR-10 dataset, our approach has a better instance segmentation performance and more accurate predicted instance masks. Moreover, the $AP_{50}$ score of HQ-ISNet is 94.6%, which has 2.9% improvements over Mask R-CNN, 3.3% gains over Mask Scoring R-CNN, and 2.3% improvements over Cascade Mask R-CNN. The HQ-ISNet attains a 74.2% $AP_{75}$, which achieves a gain of 11.4% over Mask R-CNN, 9.3% over Mask Scoring R-CNN, and 7.6% over Cascade Mask R-CNN. It has been established that the mask segmentation will be better and more precise than the other advanced approaches for instance segmentation on the NWPU VHR-10 dataset. The performance of small, medium, and large targets has also been greatly improved on the NWPU VHR-10 dataset according to $AP_S$, $AP_M$, and $AP_L$ scores. Under various AP indicators, we can obtain the same performance as HTC on the NWPU VHR-10 dataset, and some indicators exceed it, such as AP.

Furthermore, the performance gain on the NWPU VHR-10 dataset is greater than the SSDD dataset. Just as for the analysis of HRFPN and ISNetV2 in Section 4.4.2 and 4.4.3, the performance of NWPU VHR-10 is better due to the influence of target type, target size distribution, etc. In conclusion, our HRFPN can maintain a high resolution and solve the problem of spatial resolution loss in FPN, and our ISNetV2 improves the mask information flow. The final feature maps are semantically richer and spatially more accurate. The final predicted instance mask is also more accurate. Consequently, it can be extrapolated that the HRFPN and ISNetV2 modules can effectively improve instance segmentation performance in remote sensing images.

**Table 8.** Comparison to other advanced methods on the NWPU VHR-10 dataset. Where "R-50" indicates ResNet-50 and "R-101" represents ResNet-101.

| Model | Backbone | Time (ms) | Model size | AP | $AP_{50}$ | $AP_{75}$ | $AP_S$ | $AP_M$ | $AP_L$ |
|---|---|---|---|---|---|---|---|---|---|
| Mask R-CNN | R-50-FPN | 61.0 | 351M | 56.2 | 90.2 | 60.7 | 40.9 | 56.6 | 61.1 |
| | R-101-FPN | 71.4 | 503M | 57.4 | 91.7 | 62.8 | 41.0 | 57.5 | 60.5 |
| Mask Scoring R-CNN | R-50-FPN | 59.9 | 481M | 57.7 | 89.9 | 63.4 | 42.0 | 58.8 | 61.6 |
| | R-101-FPN | 71.9 | 633M | 58.8 | 91.3 | 64.9 | 41.7 | 59.1 | 65.7 |
| Cascade Mask R-CNN | R-50-FPN | 81.3 | 615M | 59.8 | 91.9 | 66.6 | 45.3 | 60.0 | 67.3 |
| | R-101-FPN | 100.0 | 768M | 60.3 | 92.3 | 65.6 | 44.6 | 60.7 | 62.4 |
| Hybrid Task Cascade | R-50-FPN | 156.2 | 639M | 65.0 | 94.1 | 72.9 | 48.3 | 65.5 | 69.8 |

|  |  |  |  |  |  |  |  |  |  |
|---|---|---|---|---|---|---|---|---|---|
|  | R-101-FPN | 166.7 | 791M | 65.7 | 94.4 | 73.4 | 50.7 | 66.2 | 75.8 |
|  | HRFPN-W18 | 120.5 | 504M | 65.6 | 93.9 | 72.2 | 50.6 | 65.9 | 76.2 |
| HQ-ISNet | HRFPN-W32 | 128.2 | 662M | 65.9 | 94.2 | 72.6 | **52.1** | 66.1 | 76.6 |
|  | HRFPN-W40 | 137.0 | 792M | **67.2** | **94.6** | **74.2** | 51.9 | **67.8** | **77.5** |

It can be observed from Tables 7 and 8 that the entire performance of HQ-ISNet performs the best with a lighter computation cost and fewer parameters. Besides, our models have a better performance than Mask R-CNN and Mask Scoring R-CNN with a similar model size and computational complexity. Compared with Cascade Mask R-CNN, our models have a better performance with less computational cost and smaller model size. Additionally, the HQ-ISNet has a similar performance compared to the Hybrid Task Cascade under the same model size, but with less runtime. Therefore, our network is more efficient and practical than other advanced approaches in terms of model size and computation complexity.

In [20], HTC introduced semantic segmentation into the instance segmentation framework to obtain a better spatial context. Because semantic segmentation requires fine pixel-level classification of the whole image, it is characterized by strong spatial position information and strong discrimination ability for the foreground and background. By reusing the semantic information of this branch into the box and mask branches, the performance of these two branches can be greatly improved. However, to achieve this function, HTC needs a separate semantic segmentation label to supervise the training of semantic segmentation branches, which is difficult to implement without annotations. Therefore, under the same model size, we achieve a similar performance compared to HTC, but our method runs for a shorter time and is easier to implement.

In summary, compared with other advanced approaches, our network acquires more accurate instance masks and improves the instance segmentation performance in HR remote sensing imagery. There are two main reasons for this. One is that HRFPN fully utilizes multi-level feature maps and can maintain HR feature maps. The other is that ISNetV2 refines the mask information flow between the mask branches.

## 5. Discussion

We found that the $AP_L$ metrics in the SSDD dataset fluctuated greatly, so we calculated the number of target instances in SSDD according to the definition of large (area > $96^2$), medium ($32^2$ < area < $96^2$) and small (area < $32^2$) targets in Section 4.2. As can be observed in Figure 15, ship instances are mainly concentrated in small and medium target areas and $AP_L$ fluctuates greatly due to too few large ships in SSDD. According to the AP calculation formula [10], a small amount of missed detections and false alarms will cause huge changes in the $AP_L$ value. Because our instance segmentation method relies on detection performance, in NWPU VHR-10, the target instances are mainly concentrated in large and medium target areas, but the number of small targets is significantly larger than the number of large targets in SSDD.

In addition, we calculate the variance to discuss the uncertainty estimate of the quality metric. We train and test our model five times to calculate the variance. As can be observed from Table 9, $AP_L$ has the largest variance fluctuation in SSDD. It is known from Figure 15 that it is caused by too few large targets. In short, the variance of other indicators is relatively stable. Thus, our results are effective.

**Table 9.** Variance of the results for HQ-ISNet.

| Data set | Backbone | AP | $AP_{50}$ | $AP_{75}$ | $AP_S$ | $AP_M$ | $AP_L$ |
|---|---|---|---|---|---|---|---|
| SSDD | HRFPN-W18 | 0.072 | 0.143 | 0.373 | 0.047 | 0.077 | 47.712 |
|  | HRFPN-W32 | 0.018 | 0.003 | 0.453 | 0.033 | 0.087 | 45.837 |
|  | HRFPN-W40 | 0.073 | 0.003 | 0.208 | 0.035 | 0.357 | 106.652 |
| NWPU VHR-10 | HRFPN-W18 | 0.053 | 0.063 | 0.360 | 0.522 | 0.093 | 1.175 |
|  | HRFPN-W32 | 0.082 | 0.497 | 0.493 | 1.36 | 0.128 | 2.022 |
|  | HRFPN-W40 | 0.093 | 0.075 | 1.007 | 1.333 | 0.035 | 1.717 |

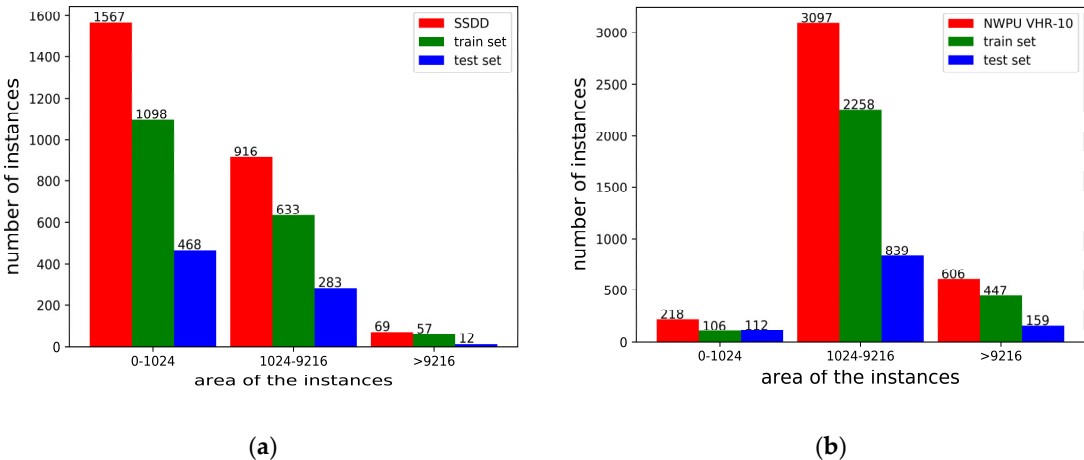

**Figure 15.** Statistical results of the instances. (**a**) SSDD; (**b**) NWPU VHR-10.

## 6. Conclusions

In this article, we put forward an instance segmentation approach based on Cascade Mask R-CNN for instance segmentation in HR remote sensing images, which is called HQ-ISNet. The HQ-ISNet adopts an HRFPN to fully utilize multi-level feature maps and maintain HR feature maps for remote sensing images' instance segmentation. Moreover, to refine mask information flow between mask branches, the instance segmentation network version 2 (ISNetV2) is proposed to promote further improvements in mask prediction accuracy. Then, we construct a new, more challenging dataset based on the SSDD and the NWPU VHR-10 dataset for remote sensing images' instance segmentation and it can be used as a benchmark for evaluating instance segmentation algorithms in the high-resolution remote sensing images. Experimental conclusions can be drawn on the SSDD and the NWPU VHR-10 dataset: (1) the HRFPN makes the predicted instance masks more accurate, which can effectively promote the instance segmentation performance of the HR remote sensing imagery; (2) the ISNetV2 is effective and promotes further improvements in mask prediction accuracy; (3) our proposed framework HQ-ISNet is effective and more accurate for instance segmentation in the remote sensing imagery than the existing algorithms. In future work, we will further study instance segmentation in SAR images.

**Author Contributions:** Software, H.S.; Methodology, H.S.; Conceptualization, H.S.; Data Curation, H.S.; Investigation, J.S.; Writing-Original Draft Preparation, H.S.; Validation, H.S., J.L. and S.L.; Resources, H.S.; Writing-Review & Editing, H.S. and C.W.; Formal Analysis, H.S.; Project Administration, X.Z.; Visualization, H.S.; Supervision, S.W.; Funding Acquisition, S.W. All authors have read and agreed to the published version of the manuscript.

**Funding:** This work was partially sponsored by the National Key R&D Program of China under Grant (2017-YFB0502700), the National Natural Science Foundation of China (61501098) and the High-Resolution Earth Observation Youth Foundation (GFZX04061502).

**Conflicts of Interest:** The authors declare no conflict of interest.

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
