# Peer review of "HQ-ISNet: High-Quality Instance Segmentation for Remote Sensing Imagery"

_remotesensing, doi:10.3390/rs12060989_

Round 1
Reviewer 1 Report
The work introduces a deep learning-based instance segmentation approach for remote sensing imagery, so called High-Quality Instance Segmentation Network (HQ-ISNet). The authors detail the main innovations in the approach, namely: the High Resolution Feature Pyramid Network (HRFPN), which serves as the backbone network for feature generation; and the Instance Segmentation Network, version 2, (ISNetV2), which is responsible for instance segmentation. Additionally, the authors evaluate the proposed approach over the SSDD and the NWPU VHR-10 datasets, originally prepared for semantic segmentation, for which they annotated individual instances of the represented classes. Finally, the authors compare the performance of the proposed approach to that obtained with state-of-the-art instance segmentation deep-learning models.
This reviewer believes the work has merits as it presents a fair degree of innovation and experimentation. However, it is my opinion that the paper needs to be improved, especially with respect to the description of the proposed approach and its alternatives, and to the experimental design and discussion of the results.
Additionally, the authors make, throughout the paper claims about the superiority of the method that are not actually supported by the presented results.
The paper also deserves an extensive revision of the written English.
In the following I list what I believe are the most important issues of the paper.
1) The authors fail to make clear some general aspects of the proposed architecture. Particularly, it seems that it is based on the previously proposed Mask R-CNN model, in which the backbone and the instance segmentation modules are substituted by HRFPN and ISNetV2, but that is never stated explicitly.
2) It is not clear how the outputs of the HRFPN are linked to the RPN. Furthermore, the architecture of the RPN is not properly described.
3) It is also not clear which are the main differences between HRFPN and FPN.
4) The description of the instance segmentation model (ISNetV2) is lacking. For the sake of clarity, the operations and components of the model need to be better explained. Particularly, the pooling operations, detection heads, RoiAlign, and the outcomes of the “detection network” are not properly described.
5) In figure 6, it seems that the outcomes of the deconvolution layers are connected to the next mask branch, which is not correct.
6) It is not clear if the first version of the instance segmentation module ISNetV1 is proposed in this work, or in [21,35].
7) As I said before, many of the claims about the superiority of the proposed approach, or of its modules, in relation to the alternatives are not supported by the results. Gains of about or less than 1% are described with terms as “clearly prove”, “greatly increased”, “significantly improve”, and so on. Actually, some of the gains are so small that one can argue about their statistical significances. In these cases, also considering the stochastic nature of deep learning models, multiple training/testing runs of the various models should have been carried out, and statistical hypothesis tests should be performed to confirm the significance of the outcomes’ differences.
8) In section 4.3, the author need to better justify the selection of the hyperparameter values (e.g., number of epochs).
9) In section 4.4.1 it is not clear to which variants of the HRFPN the results in Table 1 relate to. It is also not clear which instance segmentation model is used when ISNetV2 is not used. I suppose it is ISNetV1, but it is not clear.
10) Also in section 4.4.1, I believe it is not fair to say that the few good examples shown in Figure 9 “validates the effectiveness of our proposed approach”. The same is true for all other visual analyses.
11) Still in section 4.4.1., the results for the metric APl, do not follow the same tendency as those of the other metrics (best results are obtained when not using HRFPN), but the authors make no comment about it. The result for metric AP50 for the NWPU VHR-10 is also worse when HRFPN is used.
12) In section 4.4.2, the authors do not acknowledge that the results, when using HRFPN for APm and APl on the SSDD dataset, are actually inferior to some alternatives.
13) In section 5, tables 9 and 10, many of the results obtained with the Hybrid Task Cascade model are better than the ones obtained with the proposed approach, but the authors do not comment on that.
14) As a general remark, figures should have the same notation (e.g., type fonts) as the corresponding terms in the text.
Reviewer 2 Report
The paper presents an instance segmentation approach for high resolution remote sensing images. Multi-level feature maps are conveniently used for instance segmentation and extensive testing is presented. It is worth noting the added value of a dataset which can be used as a benchmark for different instance segmentation algorithms. The method is clearly presented and supported by a python software distribution which will be publicly distributed. As a minor change, the authors are invited to present, in a revised version of the paper, some more challenging testing examples from range-detected SAR images.
Reviewer 3 Report
Please find attached the reviewer comments.

Reviewer 4 Report
In the presented study, "HQ-ISNet: High Quality Instance Segmentation for
Remote Sensing Imagery", an advanced neural network structure is proposed namely "High Quality Instance Segmentation network" which utilizes the approach of two-stage instance detectors and combines the approaches of the Feature Pyramid Network and Cascade Mask-R-CNN with some modifications. These modifications are shown to provide an improvement in the quality of instance segmentation.
In the study, the benchmark dataset named SSDD is also presented which is supposed to be used to assess the quality of the methods for the instance segmentation tasks in high-resolution remote sensing imagery.
I find the presented manuscript written in an almost perfect manner. The text is simple yet meaningful. The introduction to the problem of instance segmentation and to the methods of instance segmentation in remote sensing imagery cover most of the known high-impact studies. The proposed method is described in a clear manner. The design of the experiment and the results of the experiment are presented clearly.
I need to mention that I would rather classify the RetinaNet [L120-121] as a two-stage detector since it has region proposals (which are not generated by a subnetwork though).
In my understanding, the text needs only minor spellchecking.
The most noticeable flaw of the presented study is the absence of the uncertainty estimates regarding the quality measures. If the authors would take a look at most of CVPR papers with CV methods or novel networks described, they would note that every quality measure mentioned with an error (uncertainty). It is crucial when one tries to compare competitive networks/approaches with close results (e.g. results in Table 7, Table 8, where the differences are about 1-2%). For example, the result of 67.7±1.2% should not be considered superior to the result of 66.5±0.7%.
I am pretty sure, that the authors already conducted series of each experiment, so calculating the variance of the quality measures should not be a huge task. Therefore I suppose it may be considered as a minor revision. for this manuscript.
Round 2
Reviewer 1 Report
This reviewer is satisfied with the improvements made by the authors in the manuscript.
I recommend an English proof reading effort to improve the text’s style in that respect.
Also, I would like to point out two minor issues:
(1) Please, give a name to the third red dashed box in Figure 2 ("instance segmentation network"?).
(2) It is still not clear to which variants of the HRFPN the results in tables 1 and 2 (data rows 2 and 4) are associated with. Please, indicate them in the tables.
Author Response
Comment 1: Please, give a name to the third red dashed box in Figure 2 ("instance segmentation network"?).
Response: Thanks for your comment. We have modified it.
Comment 2: It is still not clear to which variants of the HRFPN the results in tables 1 and 2 (data rows 2 and 4) are associated with. Please, indicate them in the tables.
Response: Thanks for your comment. In tables 1 and 2,the first line is the original baseline results; the second line is the result of HRFPN replacing FPN; the third line is the result by adding ISNetV2; the fourth line is the result obtained by the combined operation of HRFPN and ISNetV2.
Reviewer 3 Report
Please find attached the reviewer comments.

Author Response
Comment 1:High-Resolution Feature Pyramid Network (HRFPN) is previously introduced at another recent paper (Precise and Robust Ship Detection for High-Resolution SAR Imagery Based on HR-SDNet) which a co-author of this manuscript is the leading author. If there is any novelty on HRFPN different than the one described in the other paper, please clarify it.
Response: Thanks for your comment. In [10], we aim at regional-level ship detection. Then in this article, we are focusing on pixel-level instance segmentation. HRFPN fully utilizes multi-level feature maps and maintains HR feature maps. This is similar in both tasks. But our focus area is different, and the connection of HRFPN with other networks behind is also different.
Comment 2:Is there any other study focusing on instance segmentation in remote sensing? If so, it could be useful for readers to have the context if it is provided in the introduction section. Are there any specific challenges for applying instance segmentation to SAR images different than other remote sensing images?
Response: Thanks for your comment. In the remote sensing field, Mou et al. [2] came up with a novel method to perform vehicle instance segmentation of aerial images and videos obtained by UAV. Su et al. [3] introduce the precise regions of interest (RoI) pooling into the Mask R-CNN to solve the problem of loss of accuracy due to coordinate quantization in optical remote sensing images.
Compared with optical remote sensing images, SAR images are difficult to obtain. Compared with optical imaging, SAR imaging has a large difference, its characterization is not intuitive, and coherent spots and overlays during imaging are easy to interfere with target interpretation.
Comment 3:What did you change in the setup? Could you elaborate it more about the problem and how it was fixed? This could be even useful for readers. Table 1 and 2 are significantly modified in the revised version.
Response: Thanks for your comment. In Tables 1 and 2, we have added a raw baseline result. Then we replace the ISNetV2 module and HRFPN backbone to analyze the respective impacts. In addition, we take the average of 5 train and test results as our final result. In Tables 1 and 2, The first line is the original baseline results; the second line is the result of HRFPN replacing FPN; the third line is the result by adding ISNetV2; the fourth line is the result obtained by the combined operation of HRFPN and ISNetV2.